# Dysplasia Epiphysealis Hemimelica (Trevor’s Disease) in Children, Two New Cases: Diagnosis, Treatment, and Literature Review

**DOI:** 10.3390/children8100907

**Published:** 2021-10-12

**Authors:** Adelina Ionescu, Bogdan Popescu, Oana Neagu, Madalina Carp, Iulia Tevanov, Laura Balanescu, Radu Ninel Balanescu

**Affiliations:** 1Department of Pediatric Orthopedic Surgery, “Grigore Alexandrescu” Clinical Emergency Hospital for Children, 011743 Bucharest, Romania; adelina_ionescu@yahoo.com (A.I.); popescumbogdan@yahoo.com (B.P.); iulia.tevanov@gmail.com (I.T.); 2Department of Pathology, “Grigore Alexandrescu” Clinical Emergency Hospital for Children, 011743 Bucharest, Romania; oana.neagu@yahoo.com; 3Department of Pediatric Surgery, “Grigore Alexandrescu” Clinical Emergency Hospital for Children, 011743 Bucharest, Romania; laura7balanescu@yahoo.com (L.B.); radu7balanescu@yahoo.com (R.N.B.); 4Department of Pediatric Surgery and Orthopedics, University of Medicine and Pharmacy “Carol Davila”, 050474 Bucharest, Romania

**Keywords:** dysplasia epiphysealis hemimelica, Trevor’s disease, pediatric orthopedics, bone tumor

## Abstract

Dysplasia epiphysealis hemimelica (DEH), also known as Trevor’s disease, is a rare nonhereditary skeletal disorder affecting one side of the epiphyses or the epiphyses-equivalents. It is often misdiagnosed for traumatic injuries, infections, or other tumors because of the nonspecific clinical features. The diagnosis is mostly based on radiographic involvement of one half of the epiphysis displaying an overgrowth; it is hard to distinguish between DEH and osteochondroma on the gross hystopathological exam. There are few immunohistochemical markers, as well as genetic tests, for EXT1 and EXT2 gene expression that can reveal a more accurate diagnosis. No evidence of malignant changes has been reported and no hereditary transmission or environmental factor has been incriminated as an etiological factor. The natural history of the disease is continuous growth of the lesions until skeletal maturity. Without treatment, the joint might suffer degenerative modification, and the patient can develop early onset osteoarthritis. In the present paper, we report two new cases of DEH of the ankle. The aim of this paper is to consider Trevor’s disease when encountering tumoral masses in the epiphyses of pediatric patients and to present our treatment approach and results.

## 1. Introduction

Trevor’s disease or dysplasia epiphisealis hemimelica (DEH) is a disorder which implies epiphyseal overgrowth. The Greek origin of the name of the disease reveals the pattern of affecting only one half of the epiphyses from the word “hemi” meaning half and “melos” meaning limb. Most frequently it involves the knee, talus, navicular, and the first cuneiform bone [1]. The medial side of the joint, compared to the lateral side has a 2:1 incidence [2]. It was first described in 1926 by Mouchet and Berlot as “tarsomegaly” in a case of an 18-month-old boy [3]. Trevor reported eight cases of DEH in 1950 and mentions them as tarsoepiphyseal aclasis [4]. Fairbank added, 6 years later in 1956, 14 cases and gave the present name of the disease “dysplasia epiphysealis hemimelica” [3,4,5,6]. One case was discovered by Ingelrans and Lacheretz in 1953 who described it as “chondrodystrophie epiphysaire” of the medial side of the lower limb in a one year and 6-month-old boy [6]. Another case was described the same year by Donaldson as “osteochondroma of the distal femoral epiphysis” in a nine-month-old girl [6]. Since then, one or two papers per year have been written on this topic.

The incidence of Trevor’s disease is estimated to be 1:1,000,000 and affects male patients three times more often than female patients (male to female ration 3:1) [7,8]. The disease is frequently diagnosed between 2 and 8 years of age [2].

Connor et al. tried to find a genetic factor in the etiology of DEH, but there was no evidence of hereditary inheritance [9]. Hensinger et al. described, in 1974, a family with six patients suffering from DEH [10], but a more detailed analysis of these cases revealed that they suffered from a different disease. In 1974 Lenart and Aszodi reported a case of spontaneous regression [11].

Azouz et al. classified the disease in 3 clinical groups [12]:-Group 1, localized, one epiphysis is affected, often in the hindfoot, ankle or an isolated epiphysis or apophysis;-Group 2, classic, more than 1 epiphysis is affected in the same limb, same side; when localized in the foot and ankle it is named as Mouchet and Belot type;-Group 3, generalized, the entire lower limb is affected, from the pelvis to the foot

In almost 2/3 of cases more than one epiphysis is affected [13].

Fairbank described two radiological criteria of the disease [5,6]: nonunion bone growth and multiple ossification centers close to the epiphysis but unlinked to it.

In 2016, Clarke developed a new classification system based on localization of the tumor, whether it is intra or extra articular [14].

The epiphysis can be affected by multiple diseases, such as dysplasia epiphysialis multiplex, dysplasia epiphysalis punctata, dysplasia epiphysialis hemimelica, Morquio-Brailsford chondroosteodistrophy, and synovial osteochondromatosis [6]. An aggressive aneurysmal bone cyst, with involvement of the distal epiphysis of the tibia, can impair the ankle and must be differentiated from DEH [15,16,17].

There is no relationship to familial inheritance. In the case described by Donaldson in 1953, the patient had a monozygotic unaffected twin brother. Case six, described by Fairbank, by courtesy of Doctor Trevor, had a cousin with Sprengel shoulder. No link has been found between DEH and other pathologies, and there is no predilection for one side or the other of the epiphysis. Fairbank made partial or complete surgical resection of the tumoral mass with good results in every case. Particularly complex cases were those with involvement of the talus in which correction of the deformity implied a large removal of bone.

Macroscopically, the tumor is a pedunculated cauliflower-like hard mass covered by a blue-colored cartilage such as that covering the epiphysis. The surface is mostly irregular, the margins of the tumor are well defined. Microscopically, the mass is composed of hyperplastic cartilaginous cells, ununiform in size, and distributed with small dense enchondral ossification areas. None of these features are sufficient to establish the diagnosis of DEH.

The histopathology alone is unable to distinguish DEH from osteochondroma. Molecular tests for genes EXT1 and EXT2 can be used as specific tests. These tests are not recommended frequently because of their high cost, and the clinical and radiological features are generally sufficient for diagnosis. Computed Tomography, followed by 3D reconstruction, helps the surgeon choose the safest therapeutic approach [18]. Most cases involving the ankle have good prognosis after resection of the lesion, and for patients who refuse to undergo surgery, observation is recommended considering that no malignant transformation has been reported yet [19].

## 2. Case 1

In September 2020, an 8-year-old boy presented in our outpatient clinic with painful right ankle deformity and a limp. At the age of 4, a hard prominence behind the medial malleolus was observed by the parents. No traumatic event was mentioned in the medical history and family history was negative for musculoskeletal diseases. After taking an x-ray in another Orthopedic service, the child was diagnosed with a tumor of the ankle. A “wait-and-see” strategy was recommended as the child grows. In the next 3 years, the tumoral mass increased in size, deforming the hindfoot and the ankle into valgus. One year before presenting in our clinic, the child began experiencing pain after long walks or physical activity. The pain was relieved by rest. In the last year, the pain became more and more debilitating.

Physical examination revealed the presence of a tumoral mass in the right ankle, behind the medial malleolus, with hemispheric shape, about 2.5 cm in diameter. We observed a valgus deviation of the ankle and the hindfoot. We identified a painless hard mass on the posteromedial side of the right ankle, of hemispheric shape, about 2 × 3.5 cm in diameter, attached to the underlying tissue, and with no attachments to the upper planes Figure 1. The ankle range of motion was limited, being able to plantarflex about 25°, eversion of the hindfoot 30°, unable to dorsiflex the ankle (−10°), or invert the hindfoot (0°). While walking, the patient sustained his weight on the medial side of the right foot and on the entire plantar side of the left foot.

The X-ray showed multiple nonuniform ossification centers, osteochondroma-like near the right talus, anterior, posterior, and on the medial side of it, separated from the talus and the distal tibial epiphysis, deforming the articular surface on the medial side of the right tibia Figure 2.

The patient underwent further investigations, such as CT scan and MRI of the right ankle (Figure 3, Figure 4 and Figure 5). The results suggested the presence of an osteochondral mass formed by multiple centers, well separated from the talus and the tibia. CT: computed tomography, MRI: magnetic resonance imaging

The preoperative AOFAS (American Orthopaedic Foot and Ankle Society) [20] score was 50/100 points. According to Azouz classification, the patient had type 1 form of DEH.

The treatment approach was surgical. A first 12 cm incision was made on the anteromedial side of the right ankle. After protecting the tendons of the extensor muscles of the toes to the medial side, the tibiotalar articulation was opened, and the anterior part of the tumor was resected and sent for histological analysis. Another 7 cm incision was made at about 6 cm distance from the previous, behind the medial malleolus, carefully preserving the neurovascular bundle and the tibialis posterior muscle’s tendon (Figure 6). After cutting the articular capsule, the middle part of the tumor was resected and sent for anatomopathological examination (Figure 7). Between the talus and the medial malleolus, there was a large space after resection. Considering the complexity of the surgery, the high risk of skin necrosis, and the age of the patient, we decided to delay the osteotomy for another intervention (Figure 8).

A total of eight tissue fragments, between 0.5 and 3 cm in diameter, were sent for anatomopathological diagnosis. The macroscopic appearance of the tumoral fragments was a nodular polylobate solid mass. On the slide, the top of the tumor was covered by a blue-like cartilaginous cap about 0.1–0.6 cm in width. After decalcification and preparation of formalin-fixed paraffin-embedded tissue blocks the specimen was evaluated. The result indicated tissue fragments made of bone, showing chondroid proliferation on the surface, with minimal cellular atypia, with disorganized distribution and limited columnar pattern at the basal level, slightly increased cellular density, and endochondral ossification with mineralization areas (Figure 9, Figure 10, Figure 11 and Figure 12). Fibroconnective tissue was observed on the surface of chondroid proliferation (perichondrium). The result was indicative for a benign osteochondromatous structure displaying clusters of proliferative chondrocytes in a fibrillary matrix with small ossification centers and small amounts of unabsorbed calcified cartilage. Trabecular bone was covered by an irregular cartilaginous cap.

After surgery, the foot was placed in a below the knee cast for 14 days, not being allowed to walk for another 2 weeks after the removal of the cast. The patient had a prompt recovery, after taking several sessions of bioptron-light therapy and physical therapy. No recurrency of the tumor was reported until the moment of this paper.

## 3. Case 2

An eight-year-old patient presented to our clinic with deformation of the ankle. There was a visible deformity on both the posterior and lateral aspects of his left ankle. Neither the child nor parents remembered a traumatic event before the appearance of the mass. No family history of osteoarticular illnesses was mentioned. The changes in the ankle occurred almost 1 year prior to the appointment and grew progressively. The patient did not declare any kind of pain.

The X-ray of the ankle revealed a bone-like mass composed of multiple ossification centers near the left talus and the peroneal malleolus, extending posteriorly, apparently originating from the talus. There was no periosteal rection or invasion of the soft tissue surrounding the tumoral mass (Figure 13).

Based on the clinic and radiological aspects and having the recent experience with Case 1, we suspected this case to be a case of dysplasia epiphysealis hemimelica of the left talus. Considering the painless history and the personal choices of the parents, the patient is under clinical and radiological observation until present. No major changes occurred since his first registration.

## 4. Discussion

The etiology of DEH is unknown, and genetic transmission has yet to be proven. It is believed that this disease is a congenital developmental illness of the pre- or postaxial lower limb bud due to involvement of apical ectodermal cap [6]. The treatment choice varies depending on location and the severity of symptoms, and it consists of observation, surgical resection, and corrective osteotomies. When the mass is intraarticular, early surgery can determine secondary osteoarthritis, but in case of articular incongruity, early surgery should be performed to prevent damage of the articular cartilage. Kuo et al. observed recurrence of DEH and fixed deformity in intraarticular cases [21]. Contrary to dysplasia epiphysialis punctata, where the entire width of the epiphysis is involved, in DEH, only one half of the epiphysis is affected [4]. Trevor explains the pattern of DEH by the distribution of the cells and the vessels within the epiphysis [4]. In 1933, Bhosale explained that the epiphysis has a cocarde-like design in which the center of the epiphysis is formed by a bony trabecula of the ossific nucleus. Around it, there are degenerated cartilaginous cells arranged in columns, and between this layer and the mitotic annulus, which is the outermost layer, there is a zone consisting of the youngest cartilaginous cells [22]. The surface cells are old and flat in shape, and as they deteriorate, they become a component of the synovial fluid. In DEH, these cells do not become senile, and they remain capable of division forming multiple masses of cartilage with areas that later calcify.

The main complaints are pain after physical activities, swelling, deformity, limited motion, and recurrent locking of the joint. Considering our clinical experience, one of the main clinical differential diagnoses must be made with the aggressive form of an aneurysmal bone cyst of the distal tibial with involvement of the epiphysis and evolution towards the talus [15,16,17]. A simple radiography of the ankle can be helpful. Because of the massive ossification of the hypertrophic cartilage, the joints involved are suffering degenerative changes at a high rate, generating arthrosis. On radiographic images, there are multiple independent ossification centers around the epiphyses. Differential diagnosis includes posttraumatic osseous fragment, synovial chondromatosis, osteochondroma, aneurysmal bone cyst, spur of ankle, and even chondroblastoma [23]. Most of the cases of DEH in literature were treated surgically, performing resection of the tumor and even correction of the deformity while preserving the integrity of the joint surfaces. Complications of the untreated cases are angular deformity, limb length discrepancies and early degenerative arthritis.

Gökkuş et al. reported four cases of Trevor’s disease involving the ankle in 2016, all of them being treated by complete resection of the osteochondromas with no recurrence reported, returning to their old physical activities [5].

Research was made on PubMed and Google Scholar, and the papers, written in English, about DEH or Trevor’s disease in children involving the ankle were reviewed. There were reported 28 lesions in the upper extremity, one at the spinal level, and 155 studies showed a DEH localization in the pelvis and the lower extremity, enforcing the predilection of the disease for the lower limb. Unusual sites were symphysis pubis, intercondylar notch, scapula, spine, and the sacroiliac bones. After surgery, there can be a limb overgrowth because of the stimulation of the blood flow in the area. Limb shortening may be after premature epiphyseal closure because of the disease or as a complication of surgery [5].

Based on the lesion location, Acquavia et al. [24] and Kuo [21] classified the tumor as extra or intra-articular. For extra-articular lesions, simple excision is enough with good results. In case of an intraarticular location, angular deformities may need osteotomies for correction of the deformity. First, they performed an arthroscopy in order to evaluate the articular surfaces of the joint, and if the joint has integrated the lesion, they performed only hemiepiphysiodesis to correct the deformity, leaving the lesion untouched. Skriptiz et al. [25] used a combined method of treatment, performing hemiepiphysiodesis and osteotomy, resulting in minimal residual deformity. A good result after osteotomy was also obtained by Nishiyama et al. [26] in the case of a 12-year-old female patient, at the end of her growth period. Azouz et al. [12] indicate surgery whenever there is the need of correcting the deformity or function restoration, but he also mentioned the chance local vasculature stimulation and abnormal bone growth in severe forms. Malignant degeneration was not reported. After the discovery of such a lesion, the next step is to look for other sites in which the epiphyses might be involved.

Type 2 group is frequently encountered, and the medial side is twice more involved than the lateral side [12]. To reveal type 2 or 3 in patients, bone scintigraphy can detect multiple locations of the lesions.

DEH disease is a complex disease, and because of its rarity, the diagnosis and treatment are often delayed. Strujis suggests that resection is justified anytime the pain or the deformity limits the range of motion and interferes with the normal activity of the patient [19]. In our first case, the main origin of the osteochondral centers was the talus. Complete resection of the pathologic tissue is essential to limit the chances of recurrence. Limited resection can be performed if there are more surgeries in plan for the future. The surgical resection was chosen after analyzing the clinical and radiological aspects of the tumoral mass. In this case, the 3D reconstruction of the CT scan images was extremely useful in the preoperative planning of the resection and in the parents understanding the necessity of a multiple step surgery. The decision to postpone the corrective osteotomy of the ankle was made based on the remaining growth potential of the patient.

The most important differential diagnosis of DEH is osteochondroma [2], which typically occurs between 10 and 30 years of age with origin in the metaphysis of the bone, while DEH affects children of 2–8 years of age by abnormal cartilage growth and enchondral ossification of the epiphysis. Osteochondroma is one of the most common types of tumors in pediatric populations (1:50,000) [8], while DEH is a far more rare tumor in children (1:1,000,000) [7].

Both lesions are bone projections capped by cartilage but rising from different locations of the bone. Bone et al. demonstrated that exostosis genes (EXT1 and EXT2) are mutated in 90% of patients suffering from multiple hereditary exostoses but never in patients with DEH [27,28]. This suggests that the etiology of DEH is different from that of an exostosis. The mutation in EXT genes, in hereditary multiple osteochondromas, downregulates the expression of Parathyroid Hormone Like Hormone(PTHLH). An upregulation of PTHLH expression in osteochondroma signifies peripheral chondrosarcoma transformation. Therefore, Perl et al. [2] discovered that the pathogenesis of DEH involved a defect in the ability of progenitor cells to undergo programmed cell death, determining accumulation of senescent progenitor and growth plate chondrocytes in clusters, with later ossification of the lesion.

Histologically, DEH is described as an osteochondroma-like lesion, although clinically and immunohistochemically, it is a distinct entity from osteochondroma. Stevens et al. studied the differences between the two tumors and found histochemical differences between them [29]. Furthermore, they performed immunohistochemical tests for collagen type II and collagen type X to evaluate chondrocyte hypertrophy and Sox9 as a marker for proliferation of chondrocytes. Hematoxylin and eosin staining showed a different distribution of chondrocytes with different stages of maturation of cartilaginous cells in osteochondroma and less organized cells in clusters in DEH, with incomplete mineralization and little lamellar bone [30]. The cartilaginous cap in osteochondroma presents a well-organized cell distribution resembling a growth plate. Moreover, the cartilage cap in DEH showed small centers of ossification, unlike osteochondroma.

Immunohistochemical tests were performed in case 1, with samples being positive for collagen type II and proteoglycans. Our results were similar with other studies. Collagen type II was found in both osteochondroma and DEH, but the expression was high in the extracellular matrix in osteochondroma, while only a slight expression was observed surrounding the clusters of chondrocytes in DEH [29]. An increased expression was noticed under the perichondrium in DEH. Staining for collagen type X, a marker of hypertrophic chondrocytes, was absent in DEH, unlike osteochondromas [31]. Steven’s findings change Trevor’s hypothesis of apoptosis failure of hypertrophic chondrocytes and accumulation of these cells suggesting a different origin of DEH from osteochondromas. DEH presents clusters of chondrocytes with proliferative capacity, positive for Sox9 marker. Studies have shown that even the hypertrophic chondrocytes in osteochondroma showed positive results for Sox9, being able to proliferate for collagen x, therefore differentiating the two types of tumors [31] with a new hypothesis, suggesting that DEH is the result of persistent progenitor chondrocytes that present on their surface proliferative markers such as Sox9 [2].

Malignant transformation in osteochondroma is encountered in 0.5–5% of patients [32,33], and no malignant transformation of DEH has been reported to this date.

Clarke et al. [14] proposed another system of classification based on the relation between the tumor and the articular surface. This aspect influences the treatment, as there is no guideline for the treatment of this disease. The main literature consists only of case reports with limited patients and personal experiences of surgeons. This classification helps the surgeon decide whether to excise an extra articular lesion without further complication and good results, even the asymptomatic ones. As for intra-articular lesions, only the symptomatic ones should be resected because of the high rate of complications. Regular follow-up is needed until skeletal maturity, regardless of the subtype of the disease. Azzoni [34] prefers the terms juxta-articular and articular because most lesions are intracapsular, and they consider extra-articular tumors as extracapsular. They recommend an observation approach in the case of asymptomatic intra-articular involvement, as early surgery may cause osteoarthritis. Depending on the clinical expression of the tumor, asymptomatic patients may be followed-up with no other medical intervention, as no malignant transformation has been reported. In case of a painful deformity or dysfunctional form of DEH, surgical excision can be the choice in extraarticular cases by enhancing joint congruity. A more complicated surgical technique is needed in case of intraarticular lesions having the risk of reoccurrence [30].

## 5. Conclusions

Clinical, genetical, morphological, and immunohistochemical differences stand between osteochondroma and DEH. Trevor’s disease is a rare tumor that arises from the epiphysis of long bones in young children between 2 to 8 years old and has a significantly lower incidence than osteochondromas. Later studies discovered a distinctive genetic pathway related to EXT-genes and the inability to inherit DEH. Malignant transformation of DEH has not been proven yet. Clinical, radiographical, and gross histological examination cannot differentiate it from an osteochondroma. Early diagnosis and treatment can prevent a potentially disabling evolution of DEH.

## Figures and Tables

**Figure 1 children-08-00907-f001:**
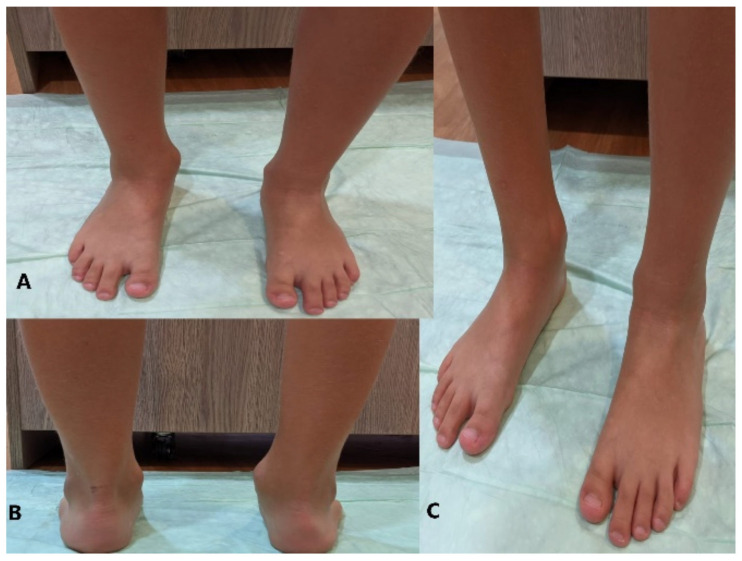
(**A**) Front view; (**B**) Posterior ankle examination; (**C**) Ankle side view.

**Figure 2 children-08-00907-f002:**
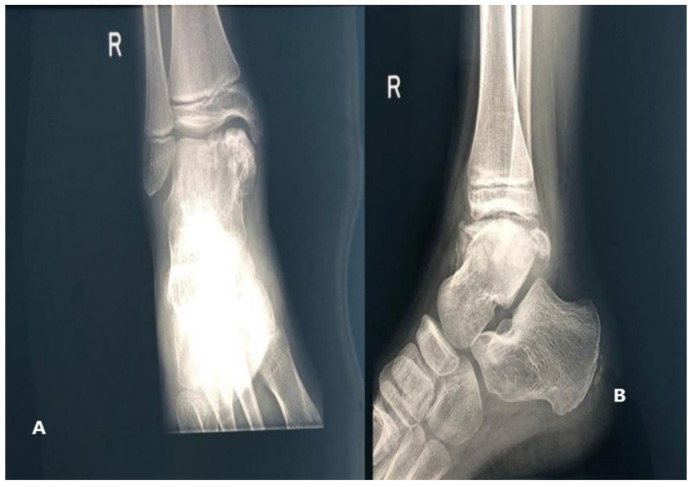
Radiologic images of Patient no 1. (**A**) Anteroposterior view (**B**) Side view.

**Figure 3 children-08-00907-f003:**
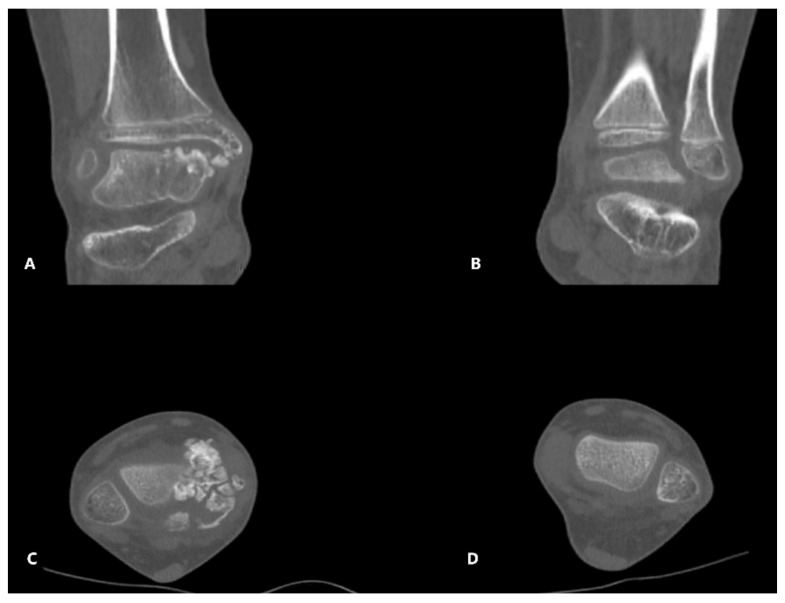
CT images—(**A**,**B**)—coronal slice, left, and right ankle; (**C**,**D**)—transverse slice—left and right ankle; images showing multiple centers of endochondral ossification near the right talus.

**Figure 4 children-08-00907-f004:**
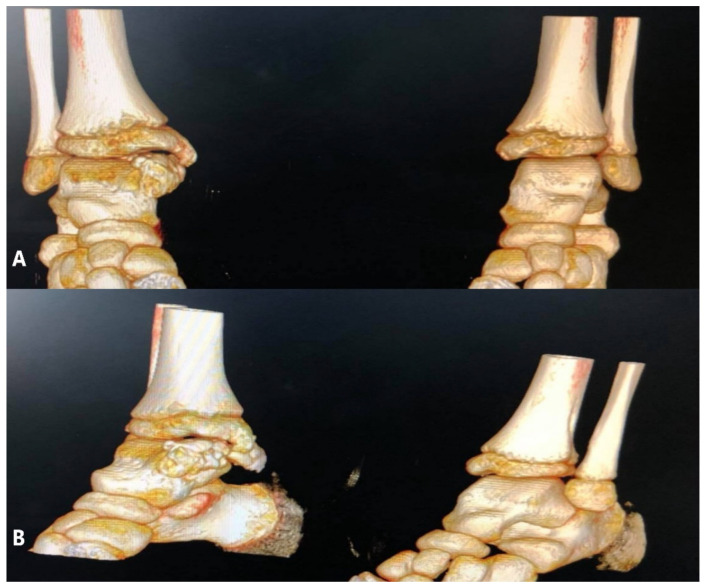
(**A**) 3D CT reconstruction of both ankles- front view; (**B**) oblique view.

**Figure 5 children-08-00907-f005:**
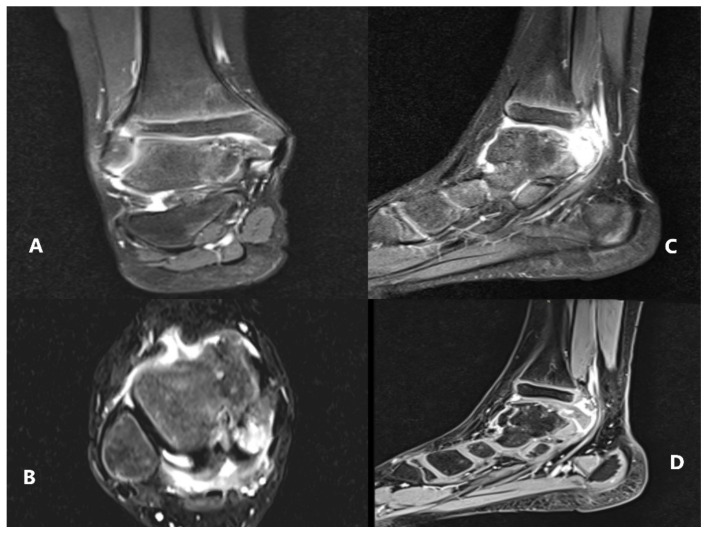
MRI scan—(**A**) coronal slice; (**B**) transverse slice; (**C**,**D**) sagittal slices demonstrating the periosseous edema and the location of the tumoral mass in relationship with the soft tissues.

**Figure 6 children-08-00907-f006:**
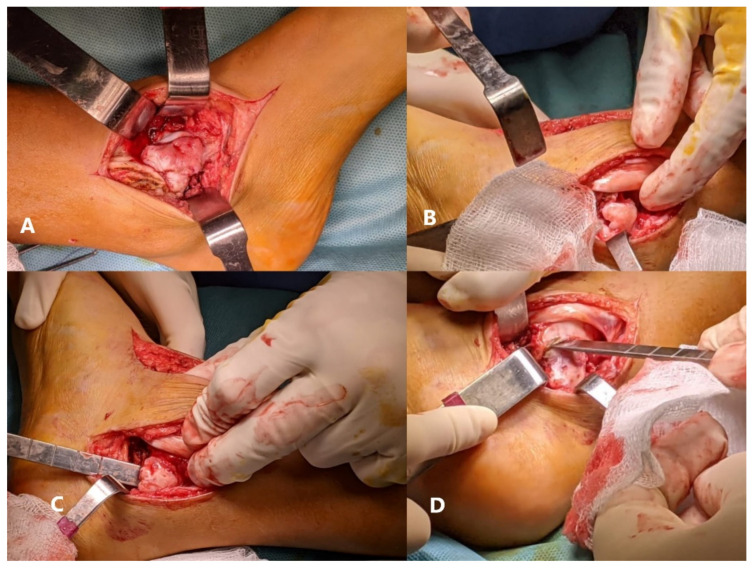
(**A**) First incision-visualization of the anterior part of the the pedunculated couliflower-like hard tumoral mass covered by a blue-colored cartilage such as that covering the epiphysis; (**B**) Second incision; (**C**) resection of the middle part of the tumor, the surface of the tumoral mass is irregular, but the margins are neat; (**D**) No macroscopical invasion in the peritumoral tissues.

**Figure 7 children-08-00907-f007:**
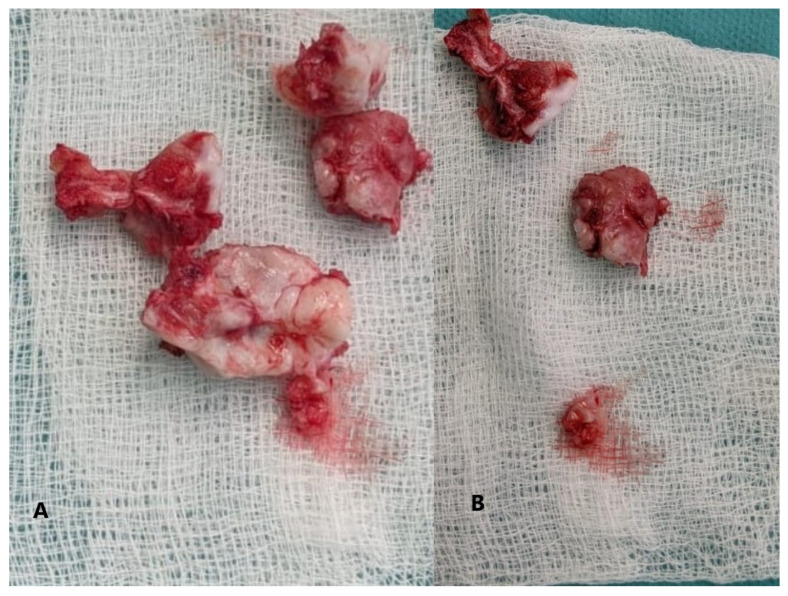
(**A**,**B**) macroscopic appearance of the resected tumor.

**Figure 8 children-08-00907-f008:**
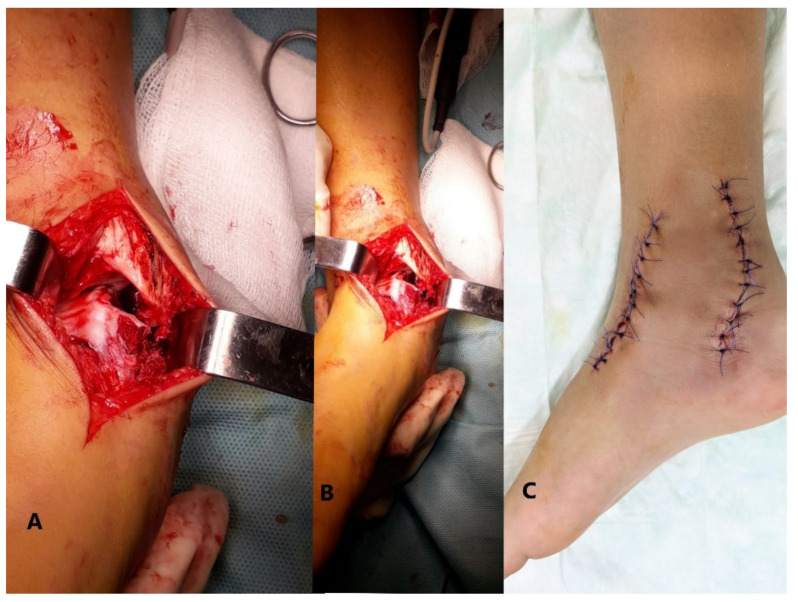
(**A**) intraoperative image after resection; (**B**) large space remaining between the talus and medial malleolus; (**C**) postoperative image.

**Figure 9 children-08-00907-f009:**
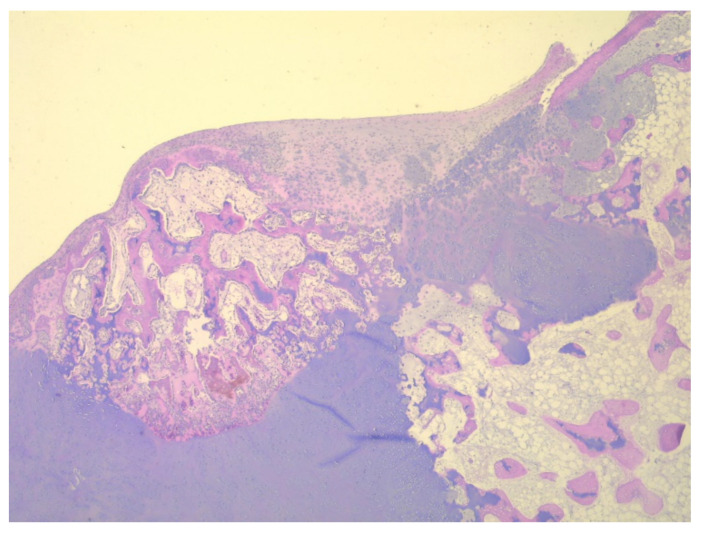
Irregular and disorganized clusters of mature chondrocytes, HE, 50×.

**Figure 10 children-08-00907-f010:**
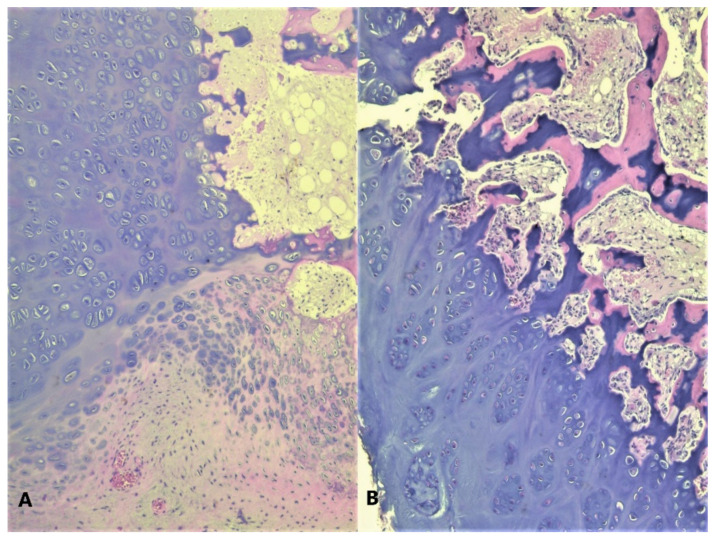
(**A**) Disorganized cell clusters with mild cytologic atypia; (**B**) Areas of cartilaginous tissue undergoing endochondral ossification in to the subjacent trabecular bone, HE, 200×.

**Figure 11 children-08-00907-f011:**
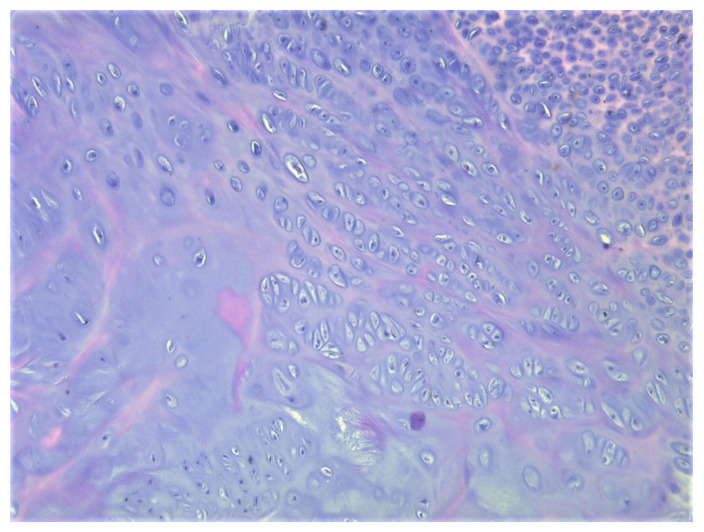
Clusters of chondrocytes with uneven cellularity and shape, HE, 200×.

**Figure 12 children-08-00907-f012:**
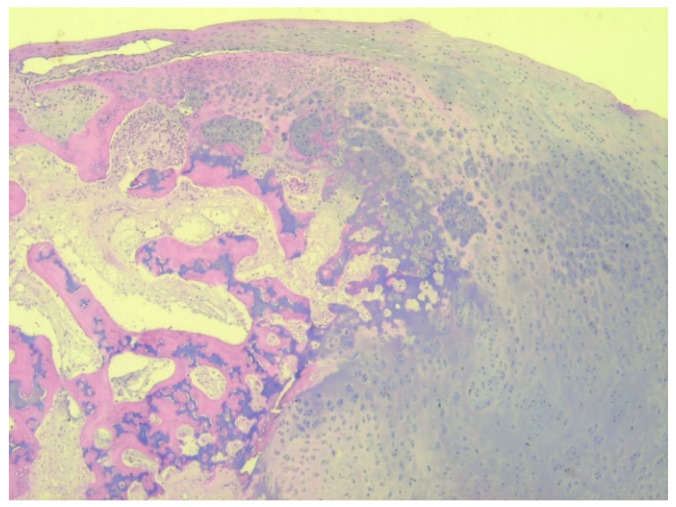
Irregular clusters of chondrocytes, ossification centers, and small amounts of unabsorbed cartilage, HE, 50×.

**Figure 13 children-08-00907-f013:**
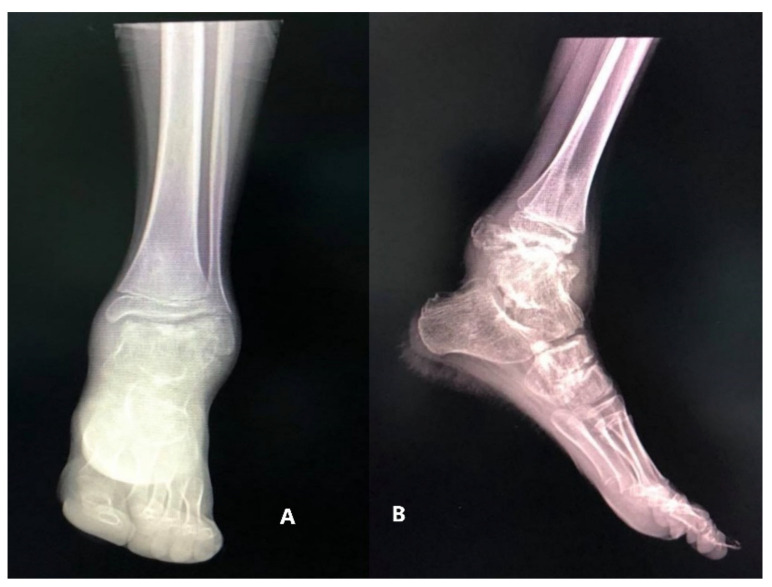
(**A**) anteroposterior view X-ray, showing a bony tumoral mass arising from the talus and extending to the peroneal malleolus; (**B**) multiple ossification centers and an overgrowth of the left talus.

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
