# Peer review of "Dysplasia Epiphysealis Hemimelica (Trevor’s Disease) in Children, Two New Cases: Diagnosis, Treatment, and Literature Review"

_children, 2021, doi:10.3390/children8100907_

Round 1

Reviewer 1 Report

It's a very well written manuscript that deal with a very rare and controversial matter. The problem is that the systematic review it's not clarified in the paper. A part with "materials and methods" it's needed to look after the methods of review (not just stating them in the discussion) and, also, a table with all the studies included in the review with cases and description of them will be very useful for understanding. Otherwise is not "systematic" bust just "review" and the title must be changed.

Author Response

Thank you for taking the time to review our paper.

Thank for your comments. We started this paper with the intension of doing a systematic review but for now we want to publish just a case presentation and a review. Thank for pointing out this aspect of our paper. We modified the title to: Dysplasia Epiphysealis Hemimelica (Trevor's Disease) in children, two new cases: diagnosis, treatment, and literature review.

We intend in the future to submit a systematic review on this subject.

We revised our English and corrected some typing mistakes.

We thank you for your careful and thorough reading of this manuscript and for the constructive suggestions which helped to improve the quality of this manuscript and helped us better define our work.

Reviewer 2 Report

dear authors
your articles is clear and interesting article
Well documented with clear images the first case. The second case is useful to underline the possibility of an initial non-surgical approach.
are lines 208 and 209 a kind of instruction from the authors as a reminder to themselves?
(This section may be divided by subheadings. It should provide a concise and precise 208
description of the experimental results, their interpretation, as well as the experimental 209)

Author Response

Thank you for taking the time to review our paper.

Thank you for your comments regarding this article and we appreciate the positive feedback.

The line 208 and 209 are an error, we apologize for this. That lines were a reminder to modify something in the discussion section.

We deleted line 208, 209 and 210 because these lines had no connection with the text.

We revised our English and corrected some typing mistakes.

We hope that these revisions are sufficient to make our manuscript suitable for publication and thank you for pointing out our mistake.

Round 2

Reviewer 1 Report

I appreciate the intent for a systematic review in future. Looking forward for it.